Corrected: Publisher correction

# Control of laser plasma accelerated electrons for light sources

T. André[1,2], I.A. Andriyash[1], A. Loulergue[1], M. Labat[1], E. Roussel[1,3], A. Ghaith[1,2], M. Khojoyan[1], C. Thaury[4], M. Valléau[1], F. Briquez[1], F. Marteau[1], K. Tavakoli[1], P. N'Gotta[1], Y. Dietrich[1], G. Lambert[4], V. Malka[4,5], C. Benabderrahmane[1], J. Vétéran[1], L. Chapuis[1], T. El Ajjouri[1], M. Sebdaoui[1], N. Hubert[1], O. Marcouillé[1], P. Berteaud[1], N. Leclercq[1], M. El Ajjouri[1], P. Rommeluère[1], F. Bouvet[1], J.-P. Duval[1], C. Kitegi[1], F. Blache[1], B. Mahieu[4], S. Corde [4], J. Gautier[4], K. Ta Phuoc[4], J.P. Goddet[4], A. Lestrade[1], C. Herbeaux[1], C. Évain[3], C. Szwaj[3], S. Bielawski[3], A. Tafzi[4], P. Rousseau[4], S. Smartsev[4,5], F. Polack[1], D. Dennetière[1], C. Bourassin-Bouchet[1], C. De Oliveira[1] & M.-E. Couprie [1,2]

With gigaelectron-volts per centimetre energy gains and femtosecond electron beams, laser wakefield acceleration (LWFA) is a promising candidate for applications, such as ultrafast electron diffraction, multistaged colliders and radiation sources (betatron, compton, undulator, free electron laser). However, for some of these applications, the beam performance, for example, energy spread, divergence and shot-to-shot fluctuations, need a drastic improvement. Here, we show that, using a dedicated transport line, we can mitigate these initial weaknesses. We demonstrate that we can manipulate the beam longitudinal and transverse phase-space of the presently available LWFA beams. Indeed, we separately correct orbit mis-steerings and minimise dispersion thanks to specially designed variable strength quadrupoles, and select the useful energy range passing through a slit in a magnetic chicane. Therefore, this matched electron beam leads to the successful observation of undulator synchrotron radiation after an 8 m transport path. These results pave the way to applications demanding in terms of beam quality.

[1] Synchrotron-SOLEIL, L'Orme des Merisiers, Saint-Aubin 91192, France. [2] Université Paris-Saclay, Paris 91190, France. [3] PhLAM, UMR CNRS 8523, Université Lille 1, Sciences et Technologies, 59655 Villeneuve d'Ascq, France. [4] LOA, École polytechnique, ENSTA ParisTech, CNRS, Université Paris-Saclay, 828 Bd des Maréchaux, 91762 Palaiseau Cedex, France. [5] Department of Physics of Complex Systems, Weizmann Institute of Science, Rehovot 761001, Israel. Correspondence and requests for materials should be addressed to M.-E.C. (email: couprie@synchrotron-soleil.fr)

The capacity of plasma waves to produce and sustain extremely strong electric fields gave rise to a high interest for plasma-based electron acceleration[1]. In the past decade, the concept of laser wakefield acceleration (LWFA) has become a reality[2–4]. Worldwide efforts presently aim at improving LWFA performance, targeting applications, such as undulator synchrotron radiation[5,6], free electron lasers[7–9], intrinsic betatron radiation[10], ultrafast electron diffraction sources[11] and even high-energy colliders[12]. In modern LWFA schemes, a high-power femtosecond laser is focused into a gas target and resonantly drives a nonlinear plasma wave. The charge-separation fields in such waves are much larger than in the radio-frequency cavities of conventional accelerators. Being able to trap the ambient plasma electrons, plasma fields accelerate them to hundreds of megaelectron-volts over few millimetre distances[13–15]. The characteristics of the produced electron beams strongly depend on how they are injected into the accelerating plasma structures. Depending on the desired application, the LWFA can be based on self injection[2–4,16,17], on triggering local injection using an auxiliary laser pulse[18,19] or by creating sharp plasma density transitions[20–22]. The localised injection requires more complicated setups but can significantly improve beam quality in terms of energy spread and divergence. In both configurations, the ionisation injection[22–26], which uses a gas with a mixture of low and high atomic number ions, enables an improvement in the source stability[22,27], but may also lead to a higher energy spread. By guiding the laser pulse at high intensity over a longer distance, high-energy electrons are obtained[28]. Experimentally, robust operation of LWFA, with all state-of-the-art beam characteristics (multi-gigaelectron-volt energies, hundreds pico-Coulomb charge, sub-percent energy spread and sub-milliradian divergence) remains extremely challenging. Moreover, the transport of such beams holds nontrivial complications, since beams with large energy spread and divergence develop a chromatic behaviour, that leads to a dramatic growth in emittance[29–32] and consequently beam quality degradation in the transfer lines. While conventional accelerators deliver microradian divergence and per mille of energy spread beams, the quality issues of LWFA require specific electron beam manipulation in order to fit the FEL application requirements, in particular to handle the large initial divergence with conventional permanent magnet quadrupole type[5,6,33–35] or plasma-based[36–39] focusing devices, and to mitigate the energy spread with magnetic chicanes for beam decompression[40–42] or transverse gradient undulators[43]. A proper electron beam control is one of the main challenges towards the Graal of developing a compact alternative of X-ray free-electron lasers by coupling LWFA gigaelectron-volts per centimetre acceleration gradient with undulators in the amplification regime.

Here we show that the LWFA beam properties can be controlled thanks to a dedicated transfer line comprising multiple magnetic devices[44]. The variable strength quadrupoles handle beam divergence and control its chromatic focusing along the beamline. A special alignment strategy was developed for separate correction of beam dispersion and position, and mitigation of pointing fluctuations. A slit equipped magnetic chicane decompresses the beam and selects the desired energy range. We finally matched this shaped electron beam to successfully produce undulator radiation.

## Results

### Concept and configuration

A schematic of the manipulation line is shown in Fig. 1 (see Methods section for details of the equipment). The LWFA is operated in the ionisation injection mode, preferred for the sake of its simplicity and robustness to long term (several hours) operation. A Ti:Sapphire laser system delivers 1.5 J, 30 fs full width at half-maximum (FWHM) pulses focused into the supersonic jet of He and $N_2$ gas mixture. Electron beams of up to a hundred pico-Coulomb are produced in a broad energy spectrum spanning from few tens up to ~250 MeV with few milliradians divergence. This large divergence is rapidly handled via strong focusing using a first set of removable permanent magnet quadrupoles, located 5 cm downstream from the gas jet. A magnetic chicane then longitudinally stretches the beam, sorts electrons in energy and selects the energy range of interest via a removable and adjustable slit mounted in the middle of the chicane. The second set of quadrupoles matches the beam inside an in-vacuum undulator that creates a periodic magnetic field (period $\lambda_u = 18$ mm, period number $N_u = 107$). Several scintillator screens can be inserted along the line to image the electron beam in the transverse plane[45]. Transfer line components and LWFA laser are aligned within ±100 μm on the same reference axis using a laser tracker.

Passing through the undulator, the beam produces synchrotron radiation with a spectrum, which typically consists of a series of lines at the so-called resonance wavelength $\lambda_r$ and its harmonics (order $n$), given by $\lambda_r = \lambda_u(1 + K_u^2/2 + \gamma^2\theta^2)/(2n\gamma^2)$, $K_u$ being the deflection parameter proportional to the peak magnetic field and period, $\gamma$ the Lorentz factor and $\theta$ the observation angle[46]. The line is tuned for a reference energy of 176 MeV, corresponding to a $\lambda_r = 200$ nm on-axis radiation for $K_u = 1.8$ achieved for the 5 mm gap. The undulator radiation lines present a homogenous FWHM linewidth $\Delta\lambda_{\text{hom}}$ being given by $\Delta\lambda_{\text{hom}}/\lambda_r \simeq 0.9/N_u$, i.e. 0.8% in this case. But a large energy spread $\sigma_\gamma$ induces an inhomogeneous broadening $\Delta\lambda_{\sigma_\gamma}/\lambda_r \simeq 2\sigma_\gamma/\gamma$ which can completely dominate the homogeneous linewidth (by a factor 50 for 20% energy spread). The effective energy spread of electrons, and thus the radiation linewidth, can be reduced by inserting an adjustable width slit in the chicane.

### LWFA electron beam

Figure 2a shows an electron beam spectrum, measured right after the gas jet, by inserting a spectrometer when the permanent magnet quadrupoles are extracted from the beamline. It exhibits a broad energy spectrum. The vertical divergence $\sigma_z'$ is analysed by slice because of the energy dependence, and the average value is found to be 4.5 mrad root mean square (RMS) with a standard deviation of 0.3 mrad over 20 shots. The slice divergence in the energy range of 176 ± 5 MeV is 3.5 mrad RMS with a standard deviation of 0.2 mrad over 20 shots. The electron beam transverse distribution measured on the first screen (Fig. 2b) provides an average horizontal divergence $\sigma_x'$ of 12.1 mrad RMS with a standard deviation of 0.5 mrad and $\sigma_z'$ of 7.1 mrad RMS with a standard deviation of 0.2 mrad. This larger vertical divergence results from the low energy electrons that are not captured by the spectrometer. As revealed in Fig. 2c, d, significant pointing fluctuations (2.2 mrad RMS) are observed and can be attributed to the intrinsic features of the LWFA source.

### Divergence handling

A first triplet of strong permanent magnet quadrupoles, called QUAPEVA, ensures a strong focusing of the large electron beam divergence. Conventional quadrupoles—permanent magnet based for high gradient delivery—were preferred to plasma lenses for the sake of simplicity, flexibility, reliability and large possible demagnification. QUAPEVAs, unlike usual permanent magnet based focusing systems[5,6,34,47], can vary their gradient strengths by a factor 2[48–50] thanks to a special design combining a Halbach type quadrupole surrounded by rotating cylindrical magnets. They are also equipped with motorised translation plates for magnetic axis adjustment in both

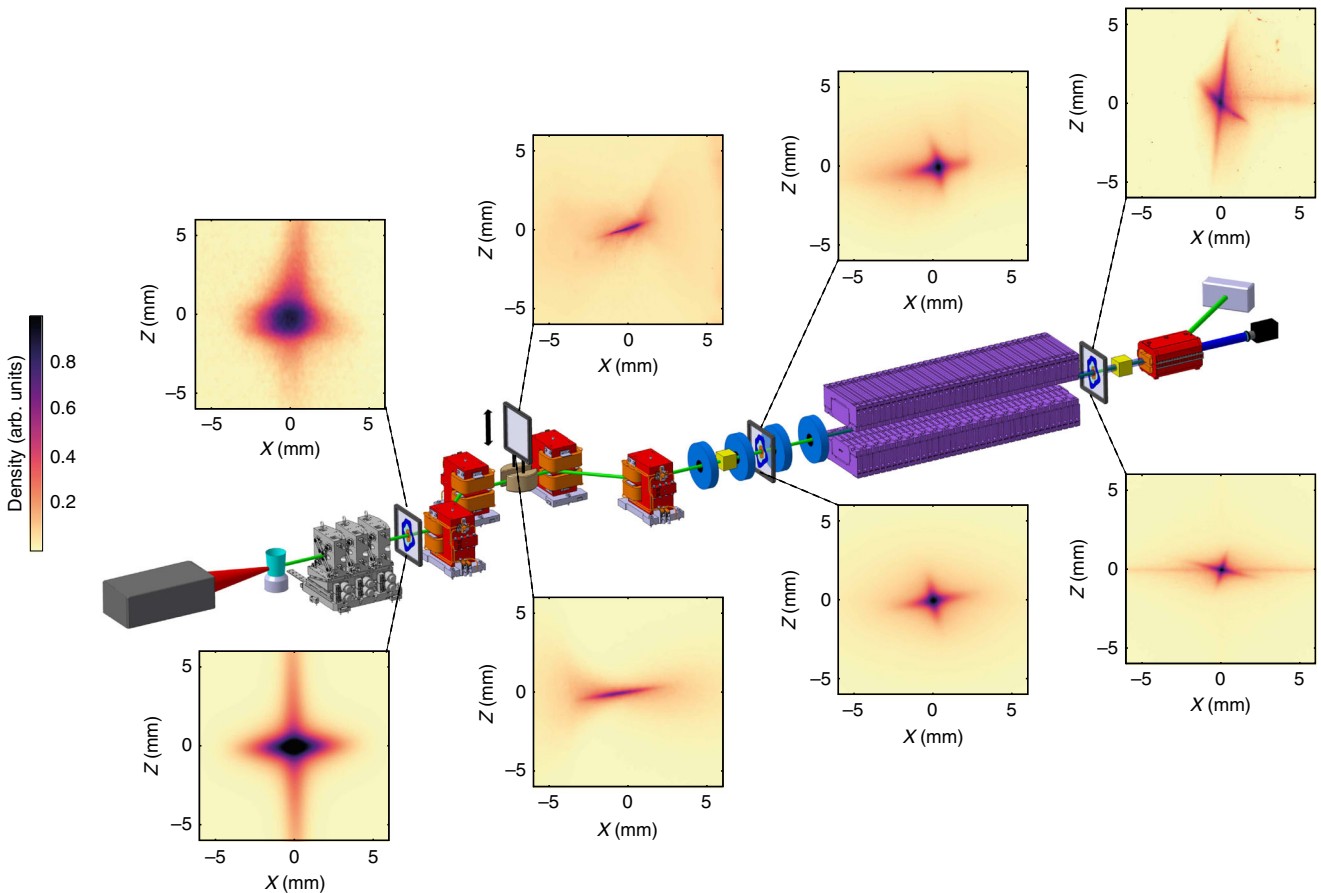

**Fig. 1** Scheme of the COXINEL manipulation line: laser hutch (grey), gas jet (cyan), removable permanent magnet quadrupoles (grey) which can be replaced by an electron spectrometer, magnetic chicane (dipole magnet in red) with a slit (brown) inserted in the middle of the chicane, electromagnetic quadrupoles (blue), undulator (purple), cavity beam position monitors (yellow), dipole dump (red), beam dump (grey) and CCD camera (black). LANEX and YAG screens for electron beam imagers. Measured (top) and simulated (down) electron beam transverse profiles (horizontal: *x* and vertical: *z* direction) along the line

transverse directions. With the QUAPEVA triplet inserted, the projected transverse size is reduced from 6 to 2 mm RMS in the horizontal plane and from 3.5 to 1 mm RMS in the vertical plane (Fig. 3a, d). Assuming an on-axis point source, the physical aperture of the QUAPEVAs let pass through more than 85% of the beam charge (Fig. 4c, d) and the transport line naturally filters the low energy electrons (Fig. 4). While the QUAPEVAs provide a strong focusing, they also permit to freeze the total-emittance growth at the exit of the triplet. The slice emittance around 176 MeV for ±5 MeV slice is typically increased from 1 to 92 (H) and 30 (V) (π.mm.mrad) due to the large initial divergence but remains then unaffected along the line. The focused beam, both measured and simulated, also exhibits a cross-like shape which results from chromatic effects (Fig. 3b, e). Indeed, the low energy electrons are focused horizontally while the high-energy ones are focused vertically (Fig. 3c, f). The transport relies on a source-to-image optics in which the focusing magnification depends on the energy range (Fig. 4a, b) (see Methods section).

**Beam alignment along the line.** To progress further down the line, while maintaining the beam quality, large orbit deviations and dispersion must be reduced to minimise the beam distortions in the undulator. Both can appear due to a misalignment of the magnetic elements with respect to the electron beam axis,

resulting either from a defect of initial position of the equipment (in particular the QUAPEVA) or from systematic and random shifts of the electron beam pointing. While in some cases this electron steering issue can be addressed using magnetised plasma guiding[37,38,51], a specific beam pointing alignment compensation (BPAC) strategy, taking advantage of the motorised translations of the QUAPEVA, is here implemented. The horizontal and vertical response matrices $\mathbf{A}_x(s)$ and $\mathbf{A}_z(s)$ of the system, at a position $s$ along the line, that link the beam position and dispersion to the transverse offset of the magnetic centre of the three QUAPEVAs, are solved (see Methods section). The proper positions of the quadrupole magnetic centres are tuned to independently minimise the transverse offset and the dispersion according to the required correction on a given screen in both planes. This procedure is illustrated in Fig. 5 for two positions along the line. In the middle of the chicane, where a strong horizontal dispersion is produced, the vertical dispersion is corrected (Fig. 5a) (tilted beam in position I rotates towards position II). A ±400 μm maximum correction of QUAPEVA transverse displacement mitigates the residual alignment errors of the quadrupoles and a change of the electrons pointing. Then, on a daily basis, adjustments remain within 10–150 μm. Further steps of BPAC include optimisation of the beam on the downstream screens before and after the undulator. The horizontal dispersion is then corrected on the screen located in front of the undulator

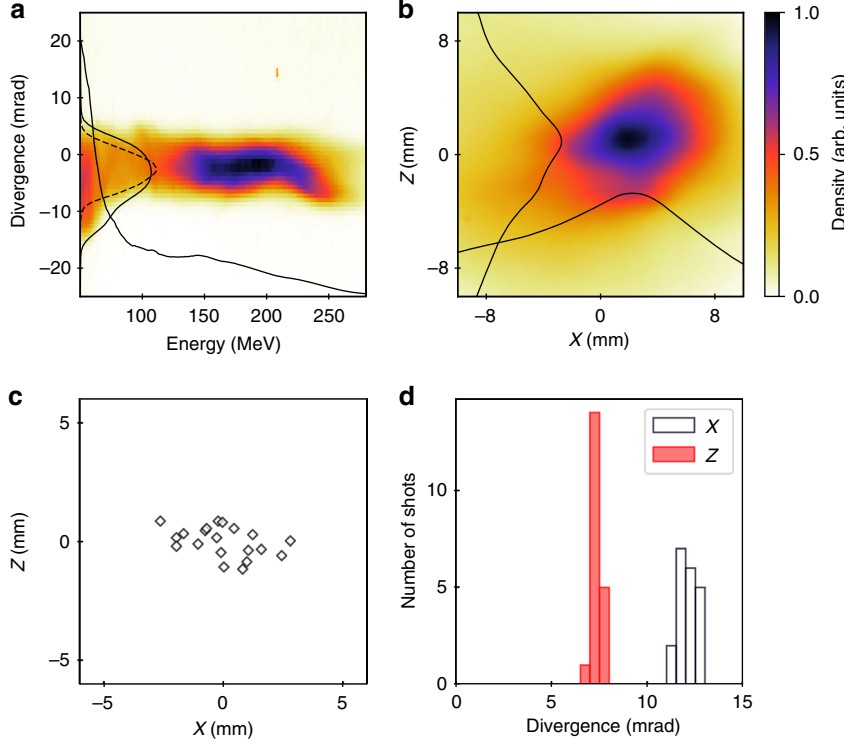

**Fig. 2** Measured LWFA beam energy and transverse distribution without QUAPEVA. **a** Measurement with the spectrometer before the first screen, with corresponding energy profile and vertical divergence evaluated by superimposing the divergence in energy slices of ±1 MeV and renormalised to the charge in the given slice (solid line), and in the case of 176 ± 5 MeV (dashed line). **b** Electron beam transverse distribution observed on the first LANEX screen located at 64 cm from the source, without spectrometer, with corresponding horizontal and vertical profiles. **c** Shot-to-shot measured pointing stability and **d** associated statistics on the electron beam sizes

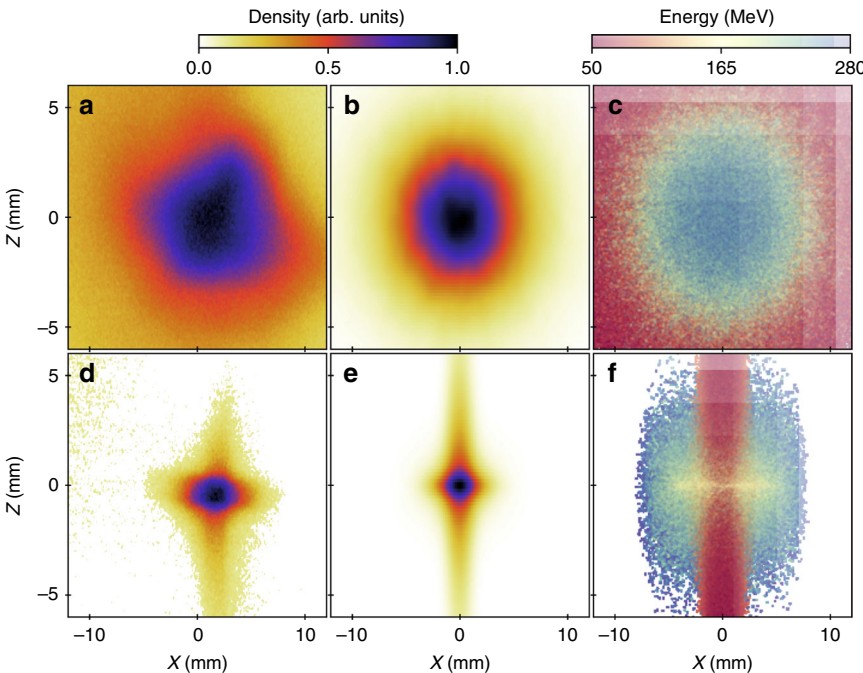

**Fig. 3** Electron beam transverse profiles observed on the first LANEX screen. Measured transverse profile without (**a**) and with (**d**) permanent magnet quadrupoles of variable strength (QUAPEVA). **b**, **e** Associated numerical simulations (see Methods section) of the transverse profile assuming a broadband energy spectrum spanning from 50 to 280 MeV and using the measured divergences and a screen resolution of 150 μm and simulated electron energy distribution in the transverse plane (**c**, **f**)

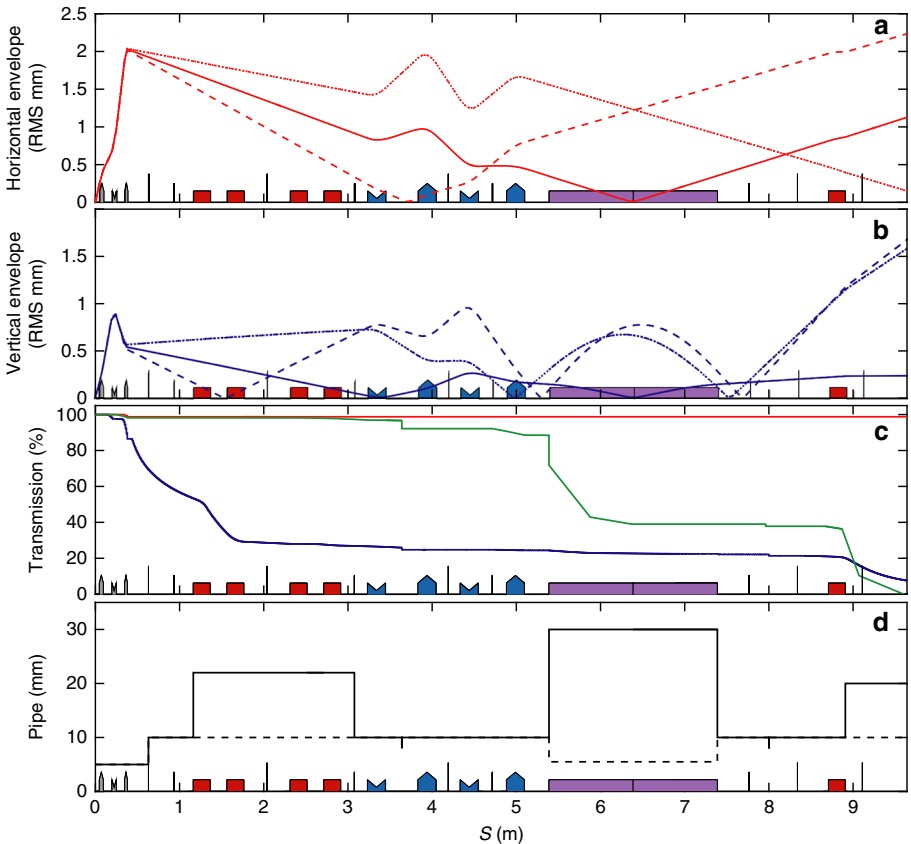

**Fig. 4** Electron beam properties along the line. **a** Horizontal and **b** vertical envelopes for 171 (dashed), 176 (solid) and 181 (dotted) MeV electron beam energies. **c** Losses along the line: 176 (red), 150 (green) MeV, spectrum from Fig. 2 (blue). **d** Horizontal (vertical) pipe diameter: dashed (solid). QUAPEVAs (grey), dipole (red), electromagnetic quadrupoles (blue), undulator (purple)

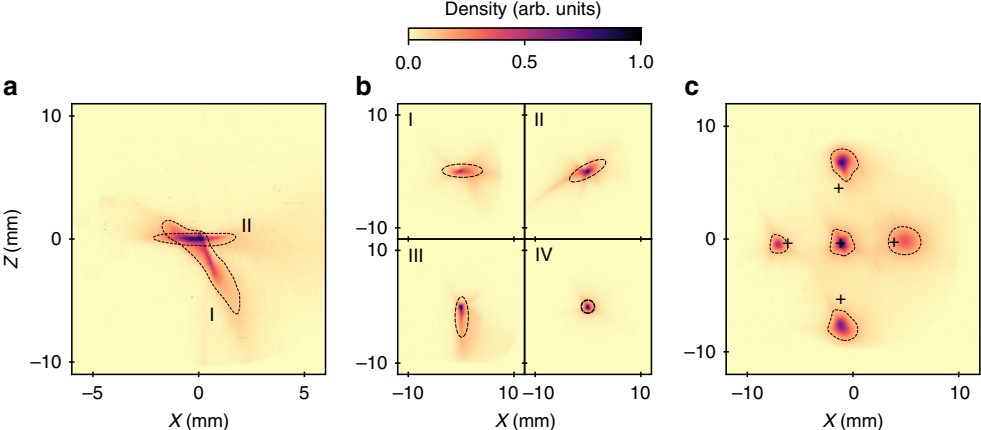

**Fig. 5** Beam pointing alignment compensation alignment method. Superimposed images with the appropriate adjustment of the QUAPEVA magnetic axis. **a** Case of screen in the middle of the chicane where the beam is horizontally dispersed, correction of the vertical dispersion (from I to II). On screen located at the undulator entrance: **b** initial beam (I), with artificial vertical dispersion introduced (II), with horizontal dispersion corrected (III), with artificial vertical dispersion removed (IV), **c** beam experimental transverse position control with respect to expected displacements from the model (black crosses)

(Fig. 5b): the initial beam (I) is artificially dispersed vertically leading to a tilted image (II) indicating the presence of horizontal dispersion, which is then suppressed by rotating the beam towards the vertical direction (III), finally the introduced vertical dispersion is removed leading to a well-focused dispersion-free centred round spot (IV). Figure 5c compares the applied position corrections of the dispersion-free electron beam from the model to the experimental measurements. BPAC enables to control the

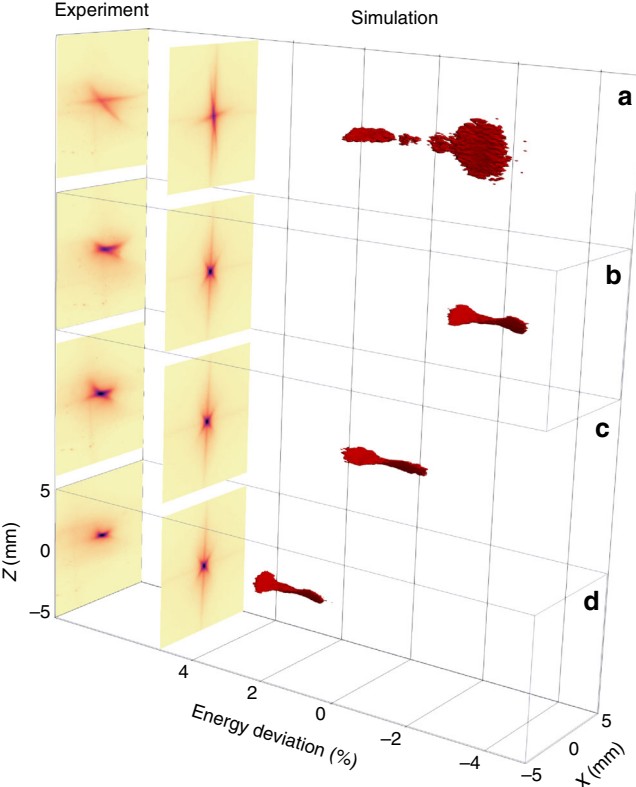

**Fig. 6** Transverse profile of the electron beam for different quadrupole strengths. Electron beam optics focusing on the screen downstream the undulator without slit, for the 176 MeV reference case. Experimental, simulated profiles and associate modelled phase-space plot **a** mismatched case, **c** well-focused case with a 1.5% correction of QUAPEVA 2, variation of the gradients of all the quadrupoles (permanent magnet and electromagnetic) by −2% (**b**) and +2% (**d**)

electron beam position and dispersion just at the exit of the QUAPEVA all along the downstream line even in presence of electron pointing and residual equipment misalignment.

**Fine tuning of the focused energies**. As a final control step, the variable gradient of the QUAPEVA allows for the fine tuning of the electron beam focusing. Minor tuning of quadrupole strength (decrease of the second QUAPEVA strength by 1.5% from the model) enables the beam shape optimisation from a cross-shaped profile (Fig. 6a) to a small round spot on the screen placed after the undulator (Fig. 6c). Numerical analysis demonstrates that the cross-shape results from electrons being focused at different planes according to their energies, as illustrated in the phase-space representations. Measurements and simulations are similar; the tilt of the cross results from a remaining QUAPEVA skew quadrupole. Furthermore, for a well-focused beam, a strength scan of all quadrupoles (QUA-PEVA and electromagnetic quadrupoles) along the line by the same ±2% amount permits to select the focused energies while keeping the electron beams well centred on the screen with the same size (Fig. 6b–d).

Following the above strategy, the electron beam is properly handled all along the line. The measured beam transverse distributions (Fig. 1, upper images) correspond indeed to the simulated ones (Fig. 1, lower images).

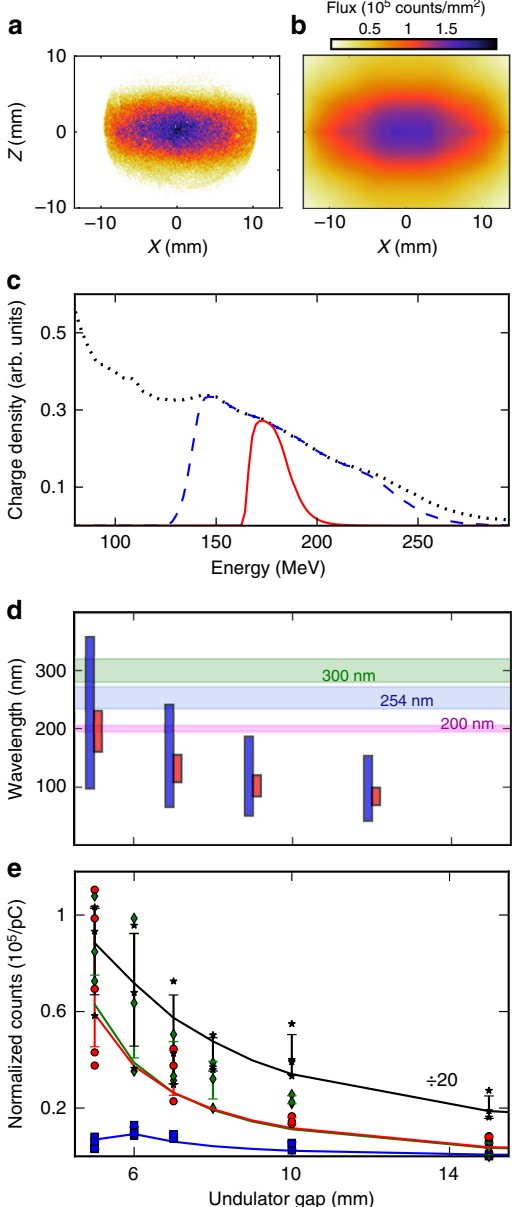

**Fig. 7** Observation of undulator radiation. Measurement (**a**) and numerical modelling (**b**) of the radiation flux density normalised to 1 pC (without slit, bandpass filters and focusing lens). **c** Spectrum measured at the exit of the electron source (dotted), and simulated at the entrance of the undulator after transport in the line (dashed), with the 4 mm slit (solid curve). **d** On-axis resonant wavelength ranges without (blue) and with (red) slit, with electrons below 10% of the maximum charge excluded and spectral FWHM bandwidth of the optical filters. **e** Total photon count measured by a camera with a lens and normalised by the beam charge black stars: without slit and bandpass filter, downscaled by a factor 10; with a 4 mm slit, red circles: 300 nm, green diamonds: 253 nm, blue squares: 200 nm filter. Error bars: mean values and deviations of acquired data sets, solid curves: numerical simulation

**Photon observation**. The shaped electron beam is suitable for the observation of the undulator synchrotron radiation with a camera installed under vacuum at the end of the line. Figure 7a, b display the transverse profiles of the radiation measured and simulated (see Methods section) without the slit: they are similar in terms of both signal level and profile shape. Simulations indicate that the physical acceptance of the beamline, defined by the vacuum

chamber geometry, naturally removes the energies below 130 MeV from the initial broad energy spectrum (Fig. 7c), leading to a reduction of the energy spread down to 30% RMS. When the 4 mm slit is inserted, the energy spread decreases to 8% RMS. A charge up to 17.4 pC, without the slit, is transported through the undulator closed at gap 5 mm. With the slit inserted, this charge drops to 5.2 pC. The resonant wavelengths corresponding to the electron beam energy range are displayed in Fig. 7d. For example, for a gap $g = 5$ mm, $\lambda_r$ spans from 98 to 358 nm without the slit, and is reduced to 161–230 nm range with the slit inserted. With larger gaps, the deflection parameter decreases as $K_u \propto \exp\left(-ag/\lambda_u + bg^2/\lambda_u^2\right)$ with $a$ and $b$ constants, and $\lambda_r$ is reduced (see Methods section). Thus, opening the gap produces two effects: the decrease of the total radiation power, $P \propto K_u^2$, and a blue-shift of the produced spectrum, $\lambda_r \propto \left(1 + K_u^2/2 + \gamma_r^2\theta^2\right)$. Figure 7e shows measurements of the integrated camera signal vs. gap above 150 nm (due to the optics system transmission), applying several spectral bandpass filters. When opening the gap, the signals decrease, both for measurements and simulations. In the case when the electron beam is not spectrally filtered (black stars), the camera receives the on-axis and the red-shifted off-axis radiation, associated with the resonant wavelengths and its harmonics. The signal follows qualitatively the dependence of the undulator total power, decreasing as the intensity collected in the detection spectral range. The measurements with the bandpass filtered inserted (coloured markers) provide a further insight on spectral behaviour. With 253 and 300 nm filters mainly off-axis light is collected, exhibiting a similar gap dependance as the total power. Alternatively, with the 200 nm narrow-band filter on-axis (at low gaps) and off-axis radiation is seen, leading to slightly different evolutions vs. gap. At 5 mm gap the camera collects the purely on-axis 200 nm light. While at 6 mm gap, the resonant wavelength decreases, the 200 nm filter band gets the red-shifted off-axis radiation whose intensity is larger than the on-axis one[46] resulting in a maximum on the gap curve (Fig. 7e). All these features clearly demonstrate the synchrotron radiation nature of the light emitted and, according to simulations, the full number of photons per beam charge at the exit of the undulator can be estimated as $N_{ph} \approx 3 \times 10^7$ pC$^{-1}$.

## Discussion

In conclusion, we have shown that the LWFA electron beam properties can be manipulated through an adequate transport line, mitigating the performance that do not meet the one of state-of-the-art conventional accelerators for some-specific applications. This electron beam control is not restricted to LWFA, but could be of interest for various advanced accelerating techniques. These results pave the way for further merging of novel and conventional accelerator techniques towards future applications, such as the new generation of colliders, requiring stages of LWFA accelerating modules or free electron laser applications, relying on more optimised LWFA performance that are progressing very fast.

## Methods

**Electron generation**. A 60 TW, 800 nm, 30 fs FWHM Ti:Sapphire laser, managed by the Laboratoire d'Optique Appliquée team in "Salle Jaune," is focused on a $12 \times 15$ μm$^2$ FWHM spot into a supersonic jet filled with a gas mixture (99% Helium, 1% Nitrogen). The precision of the laser alignment is within 10 μm in position and 10 μrad in angle. The electron beam generation is optimised thanks to a first electron spectrometer, located 355 mm from the gas jet, consisting of a removable permanent magnet dipole (1.1 T field, 10 cm length) in conjunction with a phosphor screen imaged on a CCD camera, providing a spectral resolution varying between 2.7 and 3.8% between 50 and 280 MeV. During experiments, electrons are generated, aiming at a 176 MeV reference energy, over a broad energy range. Over 422 recorded shots during 1 month of experiments, $\sigma_x'$ (respectively $\sigma_z'$) = 12.6 (5.6) mrad RMS with a standard deviation of 1.8 (1.5) mrad are measured on the first screen. These large values mainly reflect the contribution of the low energy

electrons. The energy distribution obtained with the electron spectrometer ranges from 50 to 280 MeV, and the slice vertical divergence, deduced from the spectrometer, has typical values of 3.9 mrad at $100 \pm 5$ MeV, 3.2 mrad at $150 \pm 5$ MeV and 2.9 mrad at $200 \pm 5$ MeV. The slice horizontal divergence is deduced from the average horizontal divergence ($\sigma_x' = 12.1$ mrad), measured on the first screen without QUAPEVAs, normalised by the ratio of the slice vertical divergence measured with the spectrometer, over the average divergence measured on the first screen ($\sigma_z' = 7.1$ mrad). Typical horizontal divergence values are: 6.6 mrad at $100 \pm 5$ MeV, 5.4 mrad at $150 \pm 5$ MeV and 4.9 mrad at $200 \pm 5$ MeV.

**Equipment of the transfer line**. The equipment used in the transfer line have been designed and prepared at Synchrotron SOLEIL.

Magnetic elements: all the magnetic elements have been designed using RADIA[52] and TOSCA[53] codes. After construction, they have been measured. Their fiducial references taken on the magnetic measurement bench have been reported on the installed transfer line thanks to a FARO Vantage laser tracker.

The QUAPEVA[50], developed in the frame of a SOLEIL/Sigmaphi collaboration, consists of two parts: an inner hybrid Halbach type quadrupolar ring[54] surrounded by four cylindrical magnets screened by poles, the gradient being varied by rotation of these magnets around their axis. A gradient up to 210 T m$^{-1}$ with a 50% tunability for a bore diameter of 10.5 mm can be reached, while containing reasonably the harmonic content. The installed triplet enables an operation between 176 and 400 MeV. The gradient and multipolar systematic terms measured with three different methods (stretched wire[55] of relative precision of $6 \times 10^{-4}$, rotating coil of relative precision of $8 \times 10^{-3}$[56], pulse wire[57]) correspond to the expectations from the models. The residual excursion of the magnetic centre for the gradient adjustment method (by rotating the four cylinders of the outer quadrupole in the same direction) remained in the 100 μm range, and was compensated by embedded correction tables applied on translation stages. The gradient is controlled within 0.5% mechanical precision of magnetic cylinder repositioning, the stretched wire measurement precision being much lower.

The magnetic chicane consists of four water-cooled dipoles manufactured by SEF providing 0.55 T when powered at 150 A (Sigmaphi Electronics bipolar power supplies, 8 V, 30 ppm, calibrated at SOLEIL) for 25 mm yoke gap. The electron beam dump dipole powered at 300 A provides 0.84 T. All dipoles have been measured on a SOLEIL magnetic measurement bench including a Hall probe system (consisting of three single-axis Hall probes (FW Bell GH-701) with a 1 V T$^{-1}$ sensibility) and a rotating coil (a 20 turn 3.5 m long coil made of a 0.1 mm diameter copper wire, reproducibility of 1.5%). The field measurements of one of the dipoles at 150 A with two other systems (a Group3 Teslameter and a Metrolab PT2025 NMR system) show an agreement of 10 mT, corresponding to a relative error of 1.8%, which can be explained by the relative position and orientation of the different probes, and by the hysteresis influence.

A removable aluminium slit of variable width (up to 4 mm) is inserted, in the middle of the chicane, at $32 \pm 0.5$ mm horizontal position corresponding to the 176 MeV energy, the positioning error resulting from the mechanical tolerances of the different parts.

The electromagnetic quadrupoles, manufactured by SEF, have been measured at SOLEIL with a 0.33% relative accuracy.

The 2 m long 18 mm period cryo-ready undulator, equipped with Pr$_2$Fe$_{14}$B magnets and Vanadium Permendur pole, has been developed at SOLEIL. For simplicity of infrastructure, it is operated for the experiment at room temperature, without Nitrogen cooling. In such a case, the peak magnetic field reaches 1.11 (respectively 1.005) T at 5 (respectively 5.5) mm gap, corresponding to a resonant wavelength at 200 nm for an electron beam energy of 176 (respectively 164) MeV. At 5 mm gap, 160 MeV are resonant at 243 nm, and 170 MeV at 215 nm. The magnetic field is known at 0.2%, resulting from the Hall uncertainties of calibration. The undulator deflection parameter dependence vs. gap is fitted using first harmonic of the measured magnetic field leading to: $K_u \propto \exp\left(-ag/\lambda_u + bg^2/\lambda_u^2\right)$ with $a = 3.54 \pm 0.05$ and $b = 0.22 \pm 0.05$.

Electron and photon diagnostics: the four electron beam imagers installed along the line consist of a scintillating screen mounted on a motorised stage for insertion on the electron beam axis, of imaging optics and of a camera. The screens are back-side imaged. On the first imager (LWFA exit), a LANEX type screen protected by a 75 μm black ionised aluminium foil is used together with a pair of simple focusing lenses and a 12 bit Basler scA640 CCD camera. The magnification ratio (0.12 and 0.17, respectively in the horizontal and vertical plane) together with the screen leads to a resolution of about 150 μm. For the downstream imagers, a LANEX or YAG:Ce screen protected by a 25 μm black ionised Aluminium foil can be inserted while the imaging optics is a commercial objective (f/2 100 mm focal length ZEISS MACRO or f/2.8 105 mm focal length SIGMA MACRO) and the camera is a 16 bit ORCA Flash 4.0 cMOS from Hamamatsu. The magnification ratio (0.35 and 0.5, respectively in the horizontal and vertical plane) together with the screen grain size allow to reach a resolution of 100 μm using a LANEX and of 30 μm using a YAG:Ce.

For beam charge measurements, the line is equipped with two Turbo Integrating Current Transformers (T-ICT) from Bergoz (specified for 10 fC noise), one after the electron generation chamber and a second one at the undulator exit[45], and with two cavity beam position monitors (from SwissFEL[58]) on both sides of the undulator, operating in charge mode. The screen of the imager located after the

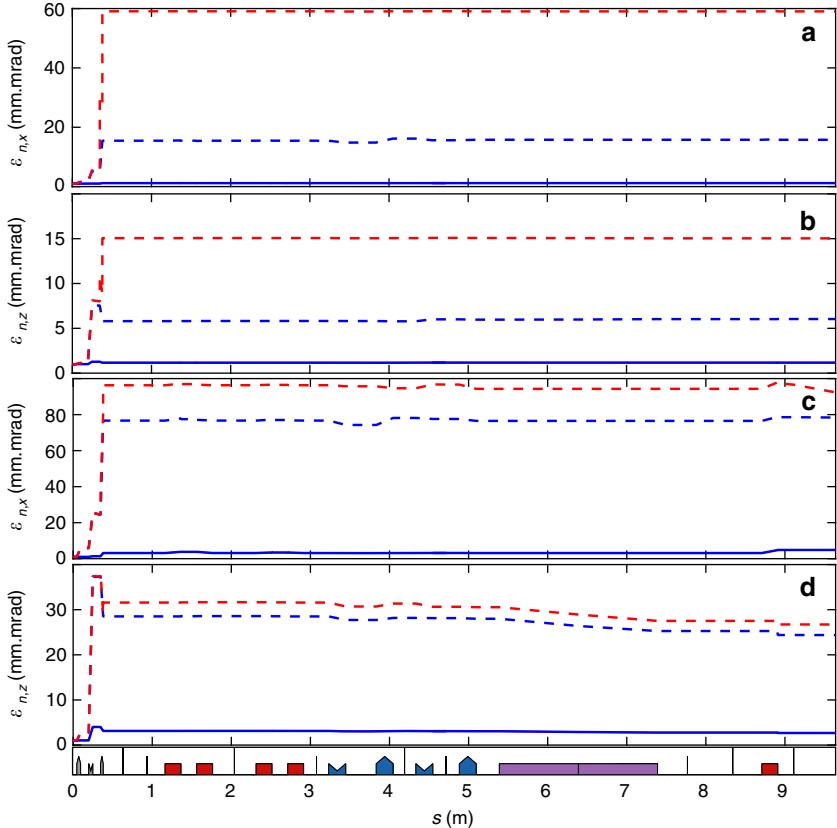

**Fig. 8** Evolution of the normalised emittance along the transport line. **a** Horizontal and **b** vertical emittance for a ±1 MeV slice and for a ±5 MeV slice **c**, **d** without (blue) and with (red) magnetic defects for a reference beam of 1 mrad divergence (solid lines) and using measured divergences from Fig. 2a (dashed lines)

undulator has been calibrated with the measurement performed with the second ICT, which has been calibrated by the supplier BERGOZ, leading to a conversion of $1.5 \times 10^7$ counts pC[−1][45]. The agreement with the absolute calibration[59–61], is found within a factor 2.

The undulator radiation was imaged with a Princeton Instruments PIXIS XO 2048 B camera, located directly under vacuum at the exit of the line, with a 75 mm fused silica biconvex lens (Newport SBX052) in front of it, limiting the detected wavelengths above 150 nm. Bandpass filters from Edmund Optics were employed for spectral selectivity (a filter centred at 200 nm (16% transmission) with a 10.2 nm FWHM width, a filter centred at 254 nm (30% transmission) with 40.8 nm FWHM width, and a filter centred at 300 nm (32% transmission) with a 46.2 nm FWHM width).

**Modelling.** Electron beam modelling: limited to the first order, the propagation of an electron through a magnetic system can be represented with a matrix formalism: $\mathbf{X}(s) = \mathbf{R} \cdot \mathbf{X}(0)$, where $\mathbf{X}(s) = (x(s), x'(s), z(s), z'(s), \zeta(s), \delta = \Delta P/P_0)$ is the six-dimensional phase-space vector that describes the electron positions and momenta at any position $s$ along the trajectory and $\mathbf{R}$ is a $6 \times 6$ matrix that represents the magnetic line with $\mathbf{R} = \mathbf{R}(n) \cdots \mathbf{R}(1) \mathbf{R}(0)$ the product of the individual transport matrix of the elements of the line (i.e. drift, quadrupoles, bendings, etc.).

The transport relies on a source-to-image optics. In the case of a monoenergetic electron beam, a first estimate of the demagnification factor due to the QUAPEVA focusing is given by $\sigma_f = \sigma_i \times R_{12}/D$ with $\sigma_f (\sigma_i)$ the focused (unfocused) beam size, $R_{12}$ the element of the transport matrix linking the position and the momentum of the electrons ($R_{12} = 0.327$ m, horizontal plane, and $R_{34} = 0.144$ m, vertical plane, $D = 0.64$ m the distance between the screen and the electron source).

The transfer line design is fitted with BETA[62] up to the second order. High order nonlinear effects and collective effects like coherent synchrotron radiation[32], are modelled using a home-made multiparticle tracking code. Similar results are obtained with ASTRA[32,63], ELEGANT[64] and OCELOT[65]. For comparison with experiment, modelling is performed with home-made code for Figs. 1, 4, 6 and 8 with ELEGANT for Figs. 3, 4 and with OCELOT for Fig. 7. A hard edge model with equivalent magnetic length is used for the QUAPEVA, and the effective focusing loss (0.5%) resulting of the longitudinal fringe field extension is accounted for experiment. Apertures of the vacuum chamber along the line are included for the

beam losses evaluation. The beamline was designed for a 100% transmission of a 1% relative energy spread beam at 176 MeV. The physical aperture of the vacuum chamber results in a charge transmission of the order of 25% over a large energy range, at the entrance of the undulator.

Electron beam parameters used for the simulations are deduced from the measurements. The electron beam distribution is taken from the charge and vertical distribution vs. energy between 50 and 280 MeV. The electron beam horizontal divergence is rescaled form spectrometer and first screen measurements. The emittance is taken constant and equal to 1.0 mm.mrad in both directions.

For the chromatic focusing optics, the quadrupole gradients are for QUAPEVA 1: +102.8 T m[−1] with skew contribution (ratio of skew gradient over normal gradient) of $+6.5 \times 10^{-3}$, for QUAPEVA 2: −101.2 T m[−1] with skew contribution of $−19.8 \times 10^{-3}$, for QUAPEVA 3: +88.17 T m[−1] and a field variation of 2% at 4-mm radius due to a dodecapole component and with skew contribution of $−18.3 \times 10^{-3}$, for QEM 1: −2.43 T m[−1], for QEM 2: +3.98 T m[−1], for QEM 3: −5.76 T m[−1], for QEM 4: +2.14 T m[−1].

For laser plasma acceleration beams presenting broad energy distribution and large divergence, the contribution of the chromatic emittance cannot be neglected, as usually done in the case of conventional accelerators. The phase-space emittance, limited to a drift space, is expressed as[30,66]: $\varepsilon_{n,x}^2 = \varepsilon_{n0,x}^2 + (\gamma s p_x'^2 \sigma_\delta)^2$ with $\varepsilon_{n0,x}$ the initial normalised emittance of the beam, $p_x$ the horizontal momentum, $\sigma_\delta$ the energy deviation and $s$ the distance from the source. For large divergence and energy spread, the second term becomes dominant. In the trace-space[67], the emittance at undulator centre in the supermatching case can be expressed up to the second order in energy deviation as[41]:

$$\varepsilon_{t,x}^2 = \varepsilon_{t0,x}^2 + \left(\frac{R_{126}}{R_{11}} \sigma_x'^2 \sigma_\delta\right)^2$$ with $\sigma_x'$ the divergence, $R_{11}$ the transport matrix elements linking the positions, $R_{126}$ the one linking the position to the transverse momentum and the energy deviation[68]. In the case of ±1 mrad divergence for which the transport line has been initially design (see Fig. 4), the trace-space emittance remains the same for the ±1 MeV slice and is contained to a factor 3 in vertical for the ±5 MeV slice. The magnetic defects (QUAPEVA remaining skew and dodecapole components) start to affect the emittance (via the first term of the equation) for divergences greater than 2 mrad. With the experimental 5.3 (3.1) mrad horizontal (vertical) divergences, the emittance grows via the chromatic term. The increase of the total emittance at the exit of the QUAPEVAs leads to a

### Table 1 Simulated electron beam properties along the transport line

| Screen location | Location | | | | | | | |
| --- | --- | --- | --- | --- | --- | --- | --- | --- |
| | Source | | First one | | Before undulator | | After undulator | |
| Slice (MeV) | ±1 | ±5 | ±1 | ±5 | ±1 | ±5 | ±1 | ±5 |
| $\sigma_x$ (μm) | 0.6 | 0.6 | 1830 | 1830 | 575 | 748 | 634 | 913 |
| $\sigma_z$ (μm) | 1.0 | 1.0 | 509 | 509 | 242 | 392 | 207 | 433 |
| $\sigma_s$ (μm) | 1.0 | 1.0 | 9 | 9 | 17 | 72 | 18 | 73 |
| $\varepsilon_{nx}$ (mm. mrad) | 1.0 | 1.0 | 58 | 93 | 58 | 92 | 58 | 92 |
| $\varepsilon_{nz}$ (mm. mrad) | 1.0 | 1.0 | 15 | 31 | 15 | 30 | 15 | 28 |

Numerical computations assuming a flat-top beam distribution, $\sigma_\delta = 1.64\%$ (±5 MeV) and $\sigma_\delta = 0.33\%$ (±1 MeV), using the measured divergences from Fig. 2a

degradation of the transverse beam brightness by a factor 92 horizontally and 30 vertically (see Table 1 and see Fig. 8), while the slice emittance (slice of ±1 MeV around 176 MeV) increases by a factor 58 (15) horizontally (vertically), indicating that the slice emittance associated to a given energy of interest is kept smaller. These chromatic effects could even be turned into an advantage for an FEL amplification application along the undulator when adopting a chromatic matching optics[41] where only a slice of a given energy is of interest.

Electron BPAC modelling: a conventional beam-based alignment (BBA) of magnetic elements would consist in varying the gradient while changing the magnetic axis of a quadrupole which would align the electron path along the magnet centre but not necessarily the electron path along the axis of the transfer line[69]. An empirical BBA technique with electron beam energy variation has been employed for single-pass free-electron lasers[70]. The proposed BPAC method consists in compensating the initial pointing of the electron beam and QUAPEVAs with respect to the line, thanks to the QUAPEVA magnetic centre resettings. In order to apply the BPAC, one needs to numerically compute the response matrix of the line linking the position and dispersion of the beam to the transverse offset of the QUAPEVAs. The transport of the beam is done numerically by applying in a first step, a transverse offset $\Delta Z_i = (\Delta x_i, \Delta z_i)$ to the magnetic centre of the QUAPEVA $i$ in consideration for the case of an ideal beam without energy dispersion (i.e. $\delta = 0$), and in a second step, by considering an electron beam with a given energy dispersion $\delta \neq 0$:

$$\mathbf{X}_{i,A}(s) = \mathbf{R}(s, \Delta\mathbf{Z}_i) \cdot \mathbf{X}(0, \delta = 0), \quad (1)$$

$$\mathbf{X}_{i,B}(s) = \mathbf{R}(s, \Delta\mathbf{Z}_i) \cdot \mathbf{X}(0, \delta \neq 0). \quad (2)$$

The induced orbit $(x_i(s), z_i(s))$ and dispersion $(D_{x,i}(s), D_{z,i}(s))$ due to the misalignment of the QUAPEVA $i$ at any position $s$ along the line are equal to:

$$x_i(s) = \langle x_{i,A}(s) \rangle, \quad (3)$$

$$z_i(s) = \langle z_{i,A}(s) \rangle, \quad (4)$$

$$D_{x,i}(s) = \frac{\langle x_{i,B}(s) \rangle - \langle x_{i,A}(s) \rangle}{\delta}, \quad (5)$$

$$D_{z,i}(s) = \frac{\langle z_{i,B}(s) \rangle - \langle z_{i,A}(s) \rangle}{\delta}, \quad (6)$$

where <> is the average over the entire electron bunch. From Eqs. (3–6), the response matrix $\mathbf{A}_{x,z}(s)$ of the transport line can be expressed as:

$$\mathbf{A}_x(s) = \begin{pmatrix} x_1(s) & x_2(s) & x_3(s) \\ D_{x,1}(s) & D_{x,2}(s) & D_{x,3}(s) \end{pmatrix}, \quad (7)$$

$$\mathbf{A}_z(s) = \begin{pmatrix} z_1(s) & z_2(s) & z_3(s) \\ D_{z,1}(s) & D_{z,2}(s) & D_{z,3}(s) \end{pmatrix}, \quad (8)$$

and the orbit $(x(s), z(s))$ and dispersion $(D_x(s), D_z(s))$ of the electron beam are

deduced by:

$$\begin{pmatrix} x(s) \\ D_x(s) \end{pmatrix} = \mathbf{A}_x(s) \cdot \begin{pmatrix} \Delta x_1 \\ \Delta x_2 \\ \Delta x_3 \end{pmatrix}, \quad (9)$$

$$\begin{pmatrix} z(s) \\ D_z(s) \end{pmatrix} = \mathbf{A}_z(s) \cdot \begin{pmatrix} \Delta z_1 \\ \Delta z_2 \\ \Delta z_3 \end{pmatrix}, \quad (10)$$

where $\Delta x_i$, $\Delta z_i$ are the QUAPEVAs transverse offset. By solving Eqs. (9, 10), one can deduce the transverse offset to apply to the QUAPEVAs to correct independently the orbit and dispersion deduced from beam positions and profiles on the given screens. The under-determined system is solved by means of a least square methods and exhibits very small residual orbit and dispersion slopes at the exit of the permanent quadrupoles making it almost valid for all screens.

Modelling of the undulator radiation: simulations of undulator radiation are performed using LWFA test particle beam with a broad energy spectrum (see dotted curve in the inset of Figs. 7c), 0.5 mm.mrad emittance, 3 mrad vertical and 5 mrad horizontal divergences. The 12,000 test particles are propagated in the undulator with $\Delta s = \lambda_u/50$ steps, using the second order Boris method[71]. Orbits are recorded and integrated to compute the spectral-angular distribution of radiation energy using classical far-field approach (see ref.[72]):

$$\frac{d\mathcal{E}}{d\Omega\,d\omega} = \frac{1}{4\pi\varepsilon_0} \frac{e^2}{4\pi^2 c} \sum_p \left| \int_{-\infty}^{\infty} dt \frac{\mathbf{n} \times (\mathbf{n} - \boldsymbol{\beta}) \times \dot{\boldsymbol{\beta}}}{(1 - n\beta)^2} e^{i\omega(t - \mathbf{n}\mathbf{r}_e/c)} \right|^2, \quad (11)$$

with $\varepsilon_0$ the vacuum permittivity. It has been checked that similar results are found with the more accurate near-field model given by[73]:

$$\mathbf{E}_\omega = \frac{1}{4\pi\varepsilon_0} \frac{ie\omega}{c} \int_{-\infty}^{\infty} dt \frac{1}{R} \left[ \boldsymbol{\beta} - \mathbf{n}\left(1 + \frac{ic}{\omega R}\right) \right] e^{i\omega(t + R/c)}. \quad (12)$$

Simulations presented in Fig. 7 are modelled in far-field approximation, as it is less sensitive to the issues related to the finite integration limits. Beam transport until the undulator is simulated with the OCELOT tracking code, which was modified to enable the large spectra beam modelling, and the orbits inside the undulator and emission is calculated with help of the dedicated modules of the home-made code CHIMERA[74].

To reproduce the measurements, the simulated spectra are treated with the optics characteristics, i.e. quantum-efficiency curve of camera, transmission of the band-pass filters and of the lens.

**Data availability**. The data that support the findings of this study are available from the corresponding authors upon reasonable request.

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

## Acknowledgements

This work was partially supported by the European Research Council for the Advanced Grants COXINEL (340015, PI: M.-E. C.) and X-Five (339128, PI: V.M.), the EuPRAXIA design study (653782) and the Fondation de la Cooperation Scientifique (QUAPEVA-2012-058T). The authors acknowledge J. Daillant, A. Nadji, P. Morin, A. Taleb, A. Thompson and A. Rousse for their support. The authors would like also to thank members of the Accelerator and Engineering Division and of the Experimental Division of SOLEIL, J. L. Lancelot and his team at Sigmaphi for the joint development of the QUAPEVA magnets.

## Author contributions

S.C., J.G., J.P.G., G.L., B.M., V.M., P. Rou., S.S., A.T., K.T.P. and C.T. worked on the laser-based electron acceleration. The line was designed by A.Lo. with M.-E.C., M.L., K.T. and modelled by T.A., I.A.A, M.K., A.Lo. and E.R. Equipments were prepared by C.B., P.B., F. Bo., F.Br., M.-E.C., Y.D., T.E.A., A.G., C.K., A.Lo., O.M., F.M., P.N., M.V. and J.V. for the magnetic elements and the undulator, T.A., C.B.-B., M.E.C., D.D., M.-E.A., N.H., G.L., A. Lo., M.L., F.P., K.T.P. and C.T. for the diagnostics. K.T. and C.D.O worked on the mechanical design and assembly, J.-P.D., C.H., P.Rom. on the vacuum, T.A., M.-E.C., M. L., A.Le., M.S. and M.V. on the alignment, T.A., I.A., F.Br., L.C., M.L, N.L., A.Lo., E.R., M.V. on the control system. T.A., I.A.A, S.B., S.C., C.E., M.-E.C., J.G., J.P.G., A.G., M.K., A.Lo., M.L., G.L., B.M., E.R., S.S., C.S., K.T.P. and C.T. worked on the experiment. Data were analysed by T.A., I.A.A., S.C., M.-E.C., M.K., A.Lo., M.L., E.R. and C.T. The paper was written by I.A.A., M.-E.C., A.Lo., M.L. and E.R.

## Additional information

**Competing interests:** The authors declare no competing interests.

