## [Peer Review File · Nature Communications]

Reviewers' comments:

Reviewer #1 (Remarks to the Author):

Comments on "Shaping of laser plasma accelerated electrons for light sources" by T. André, I. A. Andriyash, A. Loulergue, et al

The manuscript demonstrates a brilliant manipulation by characteristics of electron beams generated in laser wake fields with use of the conventional RF accelerator techniques. This is surely pioneering work deserving publication. It is well written and well presented.

However, after reading an important question remains: why such systems should be developed. Why optical systems will challenge the existing XFELs? The energy gain of GeV/cm is not a good motivation. The authors should include a paragraph with better scientific motivation for their work.

Besides, Fig.5 is not well presented especially its simulation part.

Reviewer #2 (Remarks to the Author):

This paper describes an experimental effort to study an important challenge of conditioning of the electron beam that is produced in LWFA and making it acceptable for a practical application. Authors correctly identify the main subjects of concern: large energy spread, jitter in energy, pointing and high divergence. Electron beam transport line between LWFA and radiation generation can help by filtering and demagnifying and is aim of this work. Also, ideas and need for a solution to this challenge are very timely I cannot recommend this paper for publication in its current form. Substantial rewrite is recommended.

Authors used images of electron beam to communicate how beam is conditioned by the elements of the transport line. This is done in a "hand waving" way. In contrast when optical imaging is described for different beam diagnostics authors provided all the key parameters: resolution, lens parameters, magnification... Similar level of information is needed to be communicated for the main subject of the paper - electron beam transport line.

Graphs for beam envelope for a set of different beam energies could be a useful illustration. Analysis of single lens approximation for the triplets, including chromatic and field quality induced aberrations and resulting beam brightness degradation important to be included in this paper.

"Normalized" and "arbitrary" units are used in many figures. This does not allow to see if beam size reduction after the first triplet are purely from focusing or from filtering. For example, the following statement in the paper "With the QUAPEVA triplet inserted, the projected transverse size is reduced from 6 to 2 mm RMS in the horizontal plane and from 3.5 to 1 mm RMS in the vertical plane" can be confusing without clarifying what percentage of the beam charge was transported.

Coherent synchrotron radiation (CSR) and its effect on the beam are expected to play an important role for very short bunches produced in LWFA and restrict applicability of the bending magnets. Degradation of longitudinal and transverse (in the deflection plane) phase spaces are important to quantify. CSR needs to be considered in the dipole magnets of the chicane and misaligned quadrupoles.

The paper does not address the key question of the LWFA beam conditioning transport line design: how much beam brightness is affected during beam manipulations? What was the simulations prediction and what was observed?

Reviewer #3 (Remarks to the Author):

The contribution is well constructed, showing the experiment layout and methods in detail, with each stage of the research accompanied by full simulations. While the overall output is a collection of, what could be considered, traditional beam manipulation techniques it is still a valuable result.

Abstract: It is stated that beam performance for LWFA output needs drastic improvement and the claim is that the transport line presented here can mitigate beam weaknesses. There should be a much stronger and more focussed statement here. The title targets beam shaping as the goal, the abstract writes of mitigation of beam "weakness" as a goal, and the results are using beam control and manipulation techniques. The word "shaping", which is a key word in the title, appears only once in the body of the paper. Perhaps defining this word in the context of this research would add clarity.

Results: Showing the LANEX screen data is a very an efficient way of allowing the reader to visualize the changes in beam properties. It is not clear, however, why this data is not calibrated for charge. This does not seem to be an overly complex task.

What is important here is to get an understanding of overall efficiency. This is not made explicitly clear in terms of LWFA electrons output vs final number of photons produced. A number for photons per pC is quoted but it is unclear whether this is photons per LWFA pC or per undulator entrance pC. The paper would benefit greatly from a summary table, listing the electron bunch properties at the output of the LWFA and at each of the major stages along the beam-line to the entrance of the undulator.

Discussion: This section seems to come close to a summary/conclusion, but lacks a strong concluding statement.

Methods: This section is very good. The methods are clear and references suitable.

Overall, a strong paper though some clarifications are needed.

Answer to Referee: NCOMMS-17-25575, Control of laser plasma accelerated electrons for light source applications

Reviewer n°1 (Remarks to the Author):

Comments on ‘Shaping of laser plasma accelerated electrons for light sources’ by T. André, I. A. Andriyash, A. Loulergue, et al

The manuscript demonstrates a brilliant manipulation by characteristics of electron beams generated in laser wake fields with use of the conventional RF accelerator techniques. This is surely pioneering work deserving publication. It is well written and well presented.

However, after reading an important question remains: why such systems should be developed. Why optical systems will challenge the existing XFELs? The energy gain of GeV/cm is not a good motivation. The authors should include a paragraph with better scientific motivation for their work.

The authors thank the referee for this interesting question.

It is clear that with the present electron beam performance, the LWFA cannot compete at all with the existing conventional accelerator technology, since there are still orders of magnitudes for some of the parameters (such as energy spread, divergence) and the level of reliability is quite different. However, the recent progresses of LWFA are very spectacular (more than 6 GeV reported by Leemans group at EAAC), it is worth trying to use these electrons in an application with some level of complexity. From this point of view, the FEL application can be seen as a good example to qualify the electron beam, as a step before future colliders for which roadmaps are under discussion in US and in Europe. Developing a compact alternative of XFELs by coupling LWFA of GeV/cm acceleration gradient with undulators is identified by the scientific community as the next “grand challenge”, and many groups around the world (under various funding such as many ERC grants, Moore foundation grant at Berkeley, EuPRAXIA Design Studies) are working to reach this goal, also called “the Graal”. The motivation is not to replace existing XFELs that are going to deliver ultra-bright X ray beams at high-repetition rate. For many applications, such high-repetition operation is not required and a compact and not so expensive machine delivering bright X ray beam at 10 Hz will response to the needs of a large community of users. Before producing such X ray beams, that would require GeVs electron beams, it is important to develop a strategy that will allow to reach FEL regime with electron beam parameters as those deliver by current LWFAs, in particular with an energy spread that is much larger than in conventional accelerators and to demonstrate the possibility to couple and to transport the electron beam in the undulator section by preserving their properties. The results presented here are a path towards this direction that is extremely encouraging for the future and that confirms the hope to have one day compact XFEL machines based on LWFA.

The sentence:

“In contrast to the microradian divergence and per mille of energy spread beams delivered by conventional accelerators, the quality issues of LWFA beams become crucial for some applications and stimulate developments of LWFA-compatible beam optics, e.g. conventional

permanent magnet quadrupole type [5, 6, 33–35] or plasma-based [36–39] focusing devices, magnetic chicanes for beam decompression [40–42] or transverse gradient undulators [43] to compensate the large energy spreads.”

has been modified as :

“While conventional accelerators deliver microradian divergence and per mille of energy spread beams, the quality issues of LWFA require specific electron beam manipulation in order to fit the FEL application requirements, in particular to handle the large initial divergence with conventional permanent magnet quadrupole type [5, 6, 33–35] or plasma-based [36–39] focusing devices, and to mitigate the energy spread with magnetic chicanes for beam decompression [40–42] or transverse gradient undulators [43]. A proper electron beam control is one of the main challenges in view of reaching “the Graal” of developing a compact alternative of XFELs by coupling LWFA GeV/cm acceleration gradient with undulators in the amplification regime.”

Besides, Fig.5 is not well presented especially its simulation part.

The authors modified the figure in order to improve the numerical simulation results.

The paragraph describing Figure 6 (old Figure 5):

“As a final control step, the variable gradient of the QUAPEVA allows for the fine tuning of the focusing with respect to the electron beam. Minor tuning of quadrupole strength (decrease of the second QUAPEVA strength by 1.5%) from the model enables the beam optimisation from a cross shaped profile (Fig. 5a), to a small round spot on the screen placed after the undulator (Fig. 5c). Numerical analysis demonstrates that the cross shape results from electrons being focused at different energies accordingly to their planes, as illustrated in phase space plots. Measurements and simulations are similar; the tilt of the cross results from a remaining QUAPEVA skew quadrupole component. Furthermore, the strength of all the quadrupoles (QUAPEVA and electromagnetic quadrupoles) is also varied along the line by the same $\pm 2\%$ amount (Figures 5b-d). The electron beam remains well centred on the screen while keeping the same size.”

has been modified as :

“As a final control step, the variable gradient of the QUAPEVA allows for the fine tuning of the electron beam focusing. Minor tuning of quadrupole strength (decrease of the second QUAPEVA strength by 1.5% from the model) enables the beam shape optimisation from a cross shaped profile (Fig. 6a), to a small round spot on the screen placed after the undulator (Fig. 6c). Numerical analysis demonstrates that the cross shape results from electrons being focused at different planes according to their energies, as illustrated in the phase-space representations. Measurements and simulations are similar; the tilt of the cross results from a remaining skew quadrupole component of the QUAPEVA magnets. Furthermore, when the beam is well focused, a strength scan of all quadrupoles (QUAPEVA and electromagnetic quadrupoles) along the line by the same $\pm 2\%$ amount permits to select the focused energies while keeping the electron beams well centered on the screen with the same size (Fig. 6 b-d).”

Figure 1: **Transverse profile of the electron beam for different quadrupole strengths.** Electron beam optics focusing on the screen downstream the undulator without slit, for the 176 MeV reference case. Experimental, simulated profiles and associate modelled phase space plot (a) mismatched case, (c) well-focused case with a 1.5 % correction of QUAPEVA2, variation of the gradients of all the quadrupoles (permanent magnet and electromagnetic) by -2% (b) and $+2\%$ (d).

Reviewer n°2 (Remarks to the Author):

This paper describes an experimental effort to study an important challenge of conditioning of the electron beam that is produced in LWFA and making it acceptable for a practical application. Authors correctly identify the main subjects of concern: large energy spread, jitter in energy, pointing and high divergence. Electron beam transport line between LWFA and radiation generation can help by filtering and demagnifying and is aim of this work. Also, ideas and need for a solution to this challenge are very timely I cannot recommend this paper for publication in its current form. Substantial rewrite is recommended.

Authors used images of electron beam to communicate how beam is conditioned by the elements of the transport line. This is done in a “hand waiving” way. In contrast when optical imaging is described for different beam diagnostics authors provided all the key parameters: resolution, lens parameters, magnification. . . Similar level of information is needed to be communicated for the main subject of the paper - electron beam transport line.

The authors agree with the referee that not much details were given on the electron beam transport and on the beamline performance. They have modified in consequence the article according to the referee’s suggestions, as described below.

Graphs for beam envelope for a set of different beam energies could be a useful illustration.

The authors have added a set of beam envelopes for different beam energies in a new Fig. 4, after Fig. 3 describing the effect of the first quadrupoles. The figure permits to visualize the different focusing magnification factor with respect to the electron beam energy. The sentence:

“Indeed, the low energy electrons are focused horizontally while the high energy ones are focused vertically (Fig. 3c, f). The focusing magnification depends on the energy range (see Methods).”

has been modified as:

“Indeed, the low energy electrons are focused horizontally while the high energy ones are focused vertically (Fig. 3c, f). The transport relies on a source-to-image optics in which the focusing magnification depends on the energy range (Fig. 3a, b).”

Analysis of single lens approximation for the triplets, including chromatic and field quality induced aberrations and resulting beam brightness degradation important to be included in this paper. “Normalized” and “arbitrary” units are used in many figures. This does not allow to see if beam size reduction after the first triplet are purely from focusing or from filtering.

The authors thank the referee to point out this question. Indeed, the transport has been modeled with the BETA, ASTRA [Khojoyan et al., Nucl. Instr. Meth. Phys. Res.,829, 260-264 (2016)] code and ELEGANT, including these effects. The authors have analyzed the brightness evolution with first the contribution of the transmission of the line, and second the emittance growth. The total charge transmitted along the line is strongly

Figure 2: **Electron beam properties along the line.** Horizontal (a) and vertical (b) envelopes along the line for 171 (dashed), 176 (solid) and 181 (dotted) MeV electron beam energies. (c) Losses along the line for electron beam energies of 176 MeV (red), 150 MeV (green) and for the spectrum from Fig. 2 (blue). (d) Horizontal pipe diameter (dashed) and vertical (solid). QUAPEVAs (grey), dipole (red), electromagnetic quadrupoles (blue), undulator (purple).

affected by the large energy spread distribution (as shown in the Figure below).

For example, the following statement in the paper “With the QUAPEVA triplet inserted, the projected transverse size is reduced from 6 to 2 mm RMS in the horizontal plane and from 3.5 to 1 mm RMS in the vertical plane” can be confusing without clarifying what percentage of the beam charge was transported.

The authors agree that it was not very detailed and they tried to clarify. The beam size reduction on the first screen is the result of both spectral filtering (low energies not transmitted) and focusing due to the QUAPEVAs. The simulations below show the transmission of the line taking into account the vacuum chamber aperture and the energy selection. The physical aperture of the QUAPEVAs limits the transmission of the line to 88.7%. The energy selection at the QUAPEVAs exit due to the transport optics decreases the transmission down to 86.4%. A new figure 4 (coupled to the one showing the envelopes) with the transmission of the line for different energies has been added.

Figure 3: (for answer to referee): **Electron beam losses:** For a drift space until the dipole chamber (blue), physical acceptance of the QUAPEVAs (orange) and for the nominal configuration (yellow) MeV electron beam energies.

Around the energy of interest within 176 ± 5 MeV, the transmission remains close to 100%. The degradation of the beam brightness is thus mainly dominated by the emittance growth. In the case of Laser Plasma Acceleration beams, the contribution of the chromatic emittance cannot be neglected, as usually done in the case of conventional accelerators. The authors have also added details to the electron beam modeling according to the referee's suggestion. The text (methods):

The transport relies on a source-to-image optics. A first estimate of the demagnification factor due to the QUAPEVA focusing is given by $\sigma_f = \sigma_i \times R_{12}/D$ with $\sigma_f(\sigma_i)$ the focused (unfocused) beam size, R_{12} the element of the transport matrix linking the position and the momentum of the electrons ($R_{12} = 0.327$ m, horizontal plane, and $R_{34} = 0.144$ m, vertical plane, D the distance between the screen and the electron source screen ($D = 64$ cm). The electron beam dynamics is modelled up to the second order, including high order non-linear effects, with BETA multiparticle tracking code [59]. Similar results are obtained with ASTRA [32, 60], ELEGANT [61] and OCELOT [62]. For comparison with experiment, modelling is performed with BETA for Figures 1 and 5, with ELEGANT for Figure 3 and with OCELOT for Figure 6. A hard edge model with equivalent magnetic length is used for the QUAPEVA, and the effective focusing loss (0:5%) resulting of the longitudinal fringe field extension is accounted for the experiment. Apertures of the vacuum chamber along

the line are included for the beam losses evaluation.

Electron beam parameters used for the simulations are deduced from the measurements. The electron beam distribution is taken from the charge and vertical distribution versus energy between 50 and 280 MeV. The electron beam horizontal divergence is rescaled from spectrometer and first screen measurements. The emittance is taken constant and equal to 1.0 mm.mrad in both directions. For the chromatic focusing optics, the quadrupole gradients are for QUAPEVA 1: +102.8 T/m with skew contribution (ratio of skew gradient over normal gradient) of $+6.5 \times 10^{-3}$, for QUAPEVA 2: -101.2 T/m with skew contribution of -19.8×10^{-3} , for QUAPEVA 3: +88.17 T/m with skew contribution of -18.3×10^{-3} , for QEM 1: -2.43 T/m, for QEM 2: +3.98 T/m, for QEM 3: -5.76 T/m, for QEM 4: +2.14 T/m.

have been modified to:

The transport relies on a source-to-image optics. In the case of a mono-energetic electron beam, a first estimate of the demagnification factor due to the QUAPEVA focusing is given by $\sigma_f = \sigma_i \times R_{12}/D$ with $\sigma_f(\sigma_i)$ the focused (unfocused) beam size, R_{12} the element of the transport matrix linking the position and the momentum of the electrons ($R_{12} = 0.327$ m, horizontal plane, and $R_{34} = 0.144$ m, vertical plane, $D = 0.64$ m the distance between the screen and the electron source).

The transfer line design is fitted with BETA [59] up to the second order. High order non-linear effects and collective effects like Coherent Synchrotron Radiation [XX ref martin], are modeled using a home-made multiparticle tracking code. Similar results are obtained with ASTRA [32, 60], ELEGANT [61] and OCELOT [62]. For comparison with experiment, modelling is performed with homemade code for Figures 1 and 5, with ELEGANT for Figure 3 and with OCELOT for Figure 6. A hard edge model with equivalent magnetic length is used for the QUAPEVA, and the effective focusing loss (0.5%) resulting of the longitudinal fringe field extension is accounted for the experiment. Apertures of the vacuum chamber along the line are included for the beam losses evaluation. The beamline was designed for a 100%-transmission of a 1% relative energy spread beam at 176 MeV. The physical aperture of the vacuum chamber results in a charge transmission of the order of 25% over a large energy range, at the entrance of the undulator.

Electron beam parameters used for the simulations are deduced from the measurements. The electron beam distribution is taken from the charge and vertical distribution versus energy between 50 and 280 MeV. The electron beam horizontal divergence is rescaled from spectrometer and first screen measurements. The emittance is taken constant and equal to 1.0 mm.mrad in both directions. For the chromatic focusing optics, the quadrupole gradients are for QUAPEVA 1: +102.8 T/m with skew contribution (ratio of skew gradient over normal gradient) of $+6.5 \times 10^{-3}$, for QUAPEVA 2: -101.2 T/m with skew contribution of -19.8×10^{-3} , for QUAPEVA 3: +88.17 T/m and a field variation of 2% at 4-mm radius due to a dodecapole component and with skew contribution of -18.3×10^{-3} , for QEM 1: -2.43 T/m, for QEM 2: +3.98 T/m, for QEM 3: -5.76 T/m, for QEM 4: +2.14 T/m.

For Laser Plasma Acceleration beams presenting broad energy distribution and large divergence, the contribution of the chromatic emittance cannot be neglected, as usually done in the case of conventional accelerators. The phase-space emittance, limited to a drift space, is expressed as [Floettmann et al., Phys. Rev. ST Accel. Beams, 17, 054402 (2014), Migliorati

et al., Phys. Rev. ST Accel. Beams, 16, 011302 (2013)]:

$$\epsilon_{n,x}^2 = \epsilon_{n0,x}^2 + (\gamma s p_x'^2 \sigma_\delta)^2$$

with $\epsilon_{n0,x}$ the initial normalized emittance of the beam, p_x the horizontal momentum, σ_δ the energy deviation, s the distance from the source. For large divergence and energy spread, the second term becomes dominant. In the trace-space [Floettmann et al., Phys. Rev. ST Accel. Beams, 6, 034202 (2003)], the emittance at undulator center in the supermatching case can be expressed up to the second order in energy deviation as [41]:

$$\epsilon_{t,x}^2 = \epsilon_{t0,x}^2 + \left(\frac{R_{126}}{R_{126}} \sigma_x'^2 \sigma_\delta \right)^2$$

with σ_x' is the divergence, R_{126} the transport matrix elements linking the positions, R_{126} the one linking the position to the transverse momentum and the energy deviation [ref Brown K. L. 1982SLAC report 75]. In the case of ± 1 mrad divergence for which the transport line has been initially design (see Fig. 5), the trace space emittance remains the same for the 1 MeV slice and is contained to a factor 3 in vertical for the 5 MeV slice. The magnetic defects (QUAPEVA remaining skew and dodecapole components) start to affect the emittance (via the first term of the equation) for divergences greater than 2 mrad. With the experimental 5.3 (3.1) mrad horizontal (vertical) divergences, the emittance grows via the chromatic term. The increase of the total emittance at the exit of the QUAPEVAs leads to a degradation of the transverse beam brightness by a factor 83 horizontally and 36 vertically, while the slice emittance (slice of 1 MeV around 176 MeV) only increases by a factor 20, indicating that the slice emittance associated to a given energy of interest is kept rather small. These chromatic effects could even be turned into an advantage for an FEL amplification application along the undulator when adopting a chromatic matching optics [41] where only a slice of a given energy is of interest.

The authors add a table for the referee for more details on the emittance growth for various cases.

Table 1: Table 1 (for answer to referee): **Electron beam emittances in [mm.mrad]** in the middle of the QUAPEVAs triplet and at the undulator centre.

Initial divergence $\epsilon_x = \epsilon_z = 1$ mm.mrad	176 \pm 1 MeV		176 \pm 5 MeV	
	Middle QUAPVEAs Triplet	Middle undulator	Middle QUAPVEAs Triplet	Middle undulator
$\sigma_x' = \sigma_z' = 1$ mrad no aberration	1,0 (H) 1,3 (V)	1,2 (H) 1,0 (V)	1,4 (H) 4,0 (V)	3,1 (H) 2,9 (V)
$\sigma_x' = \sigma_z' = 1$ mrad with aberration	1,0 (H) 1,3 (V)	1,2 (H) 1,0 (V)	1,4 (H) 4,0 (V)	3,1 (H) 2,9 (V)
$\sigma_x' = 5.3$ mrad, $\sigma_z' = 3.1$ mrad no aberrations	5,3 (H) 7,6 (V)	15,8 (H) 6,0 (V)	25,5 (H) 37,4 (V)	76,5 (H) 26,4 (V)
$\sigma_x' = 5.3$ mrad, $\sigma_z' = 3.1$ mrad with aberrations	6,1 (H) 8,1 (V)	59,0 (H) 15,0 (V)	25,5 (H) 37,4 (V)	94,3 (H) 28,9 (V)

The sentence in the main text:

”With the QUAPEVA triplet inserted, the projected transverse size is reduced from 6 to 2 mm RMS in the horizontal plane and from 3.5 to 1 mm RMS in the vertical plane (Fig.

Figure 4: **Evolution of the horizontal (a, c) and vertical (b, d) normalized ± 1 MeV slice (a, b) (± 5 MeV(c, d)) emittance along the transport line:** reference case of 1 mrad divergence in both plane with ideal (blue), and real with aberrations (see Methods) (red) QUAPEVAs and experimental divergences ($\sigma'_x = 5.3$ mrad, $\sigma'_z = 3.1$ mrad) (dashed lines).

3a, d).”

has been modified as:

“With the QUAPEVA triplet inserted, the projected transverse size is reduced from 6 to 2 mm RMS in the horizontal plane and from 3.5 to 1 mm RMS in the vertical plane (Fig. ??a, d). Assuming an on-axis point source, the physical aperture of the QUAPEVAs let pass through more than 85% of the beam charge (Fig. 2c,d) and the transport line naturally filters the low energy electrons (Fig. 5). While the QUAPEVAs provide a strong focusing, they also permit to freeze the total-emittance growth at the exit of the triplet. The slice emittance around 176 MeV for ± 5 MeV slice is typically increased from 1 to 94.3 (H) and 28.9 (V) (π .mm.mrad) due to the large initial divergence but remains then unaffected along the line.”

Coherent synchrotron radiation (CSR) and its effect on the beam are expected to play

an important role for very short bunches produced in LWFA and restrict applicability of the bending magnets. Degradation of longitudinal and transverse (in the deflection plane) phase spaces are important to quantify. CSR needs to be considered in the dipole magnets of the chicane and misaligned quadrupoles.

This question has been studied in ref [Khojoyan et al., Nucl. Instr. Meth. Phys. Res.,829, 260-264 (2016)], but it was not explained in the paper. Reference to the previous study has been added in the methods concerning the electron beam modelling.

The paper does not address the key question of the LWFA beam conditioning transport line design: how much beam brightness is affected during beam manipulations? What was the simulations prediction and what was observed?

Due to the broad energy distribution and large divergence, the chromatic emittance can not be neglected and leads to a significant emittance growth and consequently beam brightness reduction. In the description of the influence of the QUAPEVAs on beam quality, the authors developed the evolution of the emittance and the charge along the line. The measurement of LPA beam brightness is quite delicate. As stated by Migliorati: “The use of simplified formulas for the measurement of the normalized emittance, such as performed in [9,19,20], is therefore for laser-driven particle beams inappropriate.” . Conventional quadrupole scan does not enable to measure the beam emittance. Indeed, it requires the measurement of LPA emittance which cannot be carried out with usual methods because of the chromatic emittance contribution. In consequence, a new approach is required such as single shot emittance measurement [Weingartner et al., Phys. Rev. ST Accel. Beams, 15, 111302 (2012), Barber et al., Phys. Rev. Lett. 119, 104801] has to be used. Together with the shot-to-shot fluctuations of the electron source parameters, the implemented diagnostics, the brightness measurement is a specific work by itself, and is out of the scope of the present one.

Reviewer n°3 (Remarks to the Author):

The contribution is well constructed, showing the experiment layout and methods in detail, with each stage of the research accompanied by full simulations. While the overall output is a collection of, what could be considered, traditional beam manipulation techniques it is still a valuable result.

Abstract: It is stated that beam performance for LWFA output needs drastic improvement and the claim is that the transport line presented here can mitigate beam weaknesses. There should be a much stronger and more focussed statement here. The title targets beam shaping as the goal, the abstract writes of mitigation of beam "weakness" as a goal, and the results are using beam control and manipulation techniques. The word "shaping", which is a key word in the title, appears only once in the body of the paper. Perhaps defining this word in the context of this research would add clarity.

The authors agree that the word "shaping" was probably misused. The authors have modified the title to make it clearer: "Control of laser plasma accelerated electrons for light source applications". The sentence:

"By applying conventional accelerator techniques to laser wakefield acceleration, we have demonstrated the shaping of the electron beam longitudinal and transverse phase-spaces." has been modified to:

"By applying conventional accelerator techniques to laser wakefield acceleration, we have demonstrated the control of the electron beam longitudinal and transverse phase-spaces."

Results: Showing the LANEX screen data is a very an efficient way of allowing the reader to visualize the changes in beam properties. It is not clear, however, why this data is not calibrated for charge. This does not seem to be an overly complex task.

LANEX screens data could be used indeed to give an electron beam charge measurement provided that an absolute calibration of the screens together with their imaging systems is performed [Glinec et al., Rev. of Sci. Instr. 77, 103301 (2006), Nakamura et al., Phys. Rev. Special Topics 14, 062801 (2011), Buck et al., Rev. of Sci. Instr. vol. 81, p. 033301, (2010)]. The authors prefer thus to characterize the transport performance in terms of charge using the turbo ICTs, which have been calibrated by the supplier (BERGOZ). In the methods part, the text:

For beam charge measurements, the line is equipped with two Turbo Integrating Current Transformers (T-ICT) from Bergoz (specified for 10 fC noise), one after the electron generation chamber and a second one at the undulator exit [45], and with two cavity Beam Position Monitors (from SwissFEL [58]) on both sides of the undulator, operating in charge mode.

has been replaced by:

For beam charge measurements, the line is equipped with two Turbo Integrating Current Transformers (T-ICT) from Bergoz (specified for 10 fC noise), one after the electron generation chamber and a second one at the undulator exit [45], and with two cavity Beam Position Monitors (from SwissFEL [58]) on both sides of the undulator, operating in charge mode. The screen of the imager located after the undulator has been calibrated with the

measurement performed with the second ICT, which has been calibrated by the supplier BERGOZ, leading to a conversion of $1.5 \cdot 10^7$ counts/pC [Labat et al., *J. Synchrotron Rad.* 25, 59-67 (2018)]. The agreement with the absolute calibration [Glinec et al., *Rev. of Sci. Instr.* 77, 103301 (2006), Nakamura et al., *Phys. Rev. Special Topics* 14, 062801 (2011), Buck et al., *Rev. of Sci. Instr.* vol. 81, p. 033301, (2010)], considering the unprecise knowledge of some parameters, is found within a factor 2.

What is important here is to get an understanding of overall efficiency.

Details of the transmission of the transport line as well as degradation of the beam brightness have been given in the answer to reviewer 2. Text has been modified accordingly and a new figure with the theoretical transmission of the transport line has been added after Fig. 3 describing the effect of the first quadrupoles.

Figure 5: **Electron beam losses.** (a) Losses along the line for electron beam energies of 176 MeV (red), 150 MeV (green) and for all spectrum from Fig. 2 (blue). (b) Horizontal pipe diameter (dashed) and vertical (solid). Permanent magnet quadrupoles (grey), dipole (red), electromagnetic quadrupoles (blue), undulator (purple).

This is not made explicitly clear in terms of LWFA electrons output vs final number of photons produced. A number for photons per pC is quoted but it is unclear whether this is photons per LWFA pC or per undulator entrance pC.

The sentence:

“... the full number of photons per beam charge can be estimated as $N_{ph} \approx 3 \cdot 10^7$ per pC.”

has been modified as:

“... the full number of photons per beam charge at the exit of the undulator can be estimated as $N_{ph} \approx 3 \cdot 10^7$ per pC.”

The paper would benefit greatly from a summary table, listing the electron bunch

properties at the output of the LWFA and at each of the major stages along the beam-line to the entrance of the undulator.

The authors added in the methods before the "Electron Beam Pointing Compensation (BPAC) modelling" paragraph, a table summarizing the beam parameters at critical locations along the line. A figure with the envelope has been inserted in the main text of the paper and the summary table was included in the electron beam modelling section of the Methods part.

Table 2: Simulated electron beam properties along the transport line. Numerical computations assuming a flat-top beam distribution, $\sigma_\delta = 1,64\%$ (± 5 MeV) and $\sigma_\delta = 0,33\%$ (± 1 MeV), using the measured divergences from Fig.2a.

Properties	Location								
	Screen location		Source		First one		Before undulator		After undulator
Slice[MeV]	± 1	± 5	± 1	± 5	± 1	± 5	± 1	± 5	
$\sigma_x[\mu\text{m}]$	0.6	0.6	1830	1830	575	748	634	913	
$\sigma_z[\mu\text{m}]$	1.0	1.0	509	509	242	392	207	433	
$\sigma_s[\mu\text{m}]$	1.0	1.0	8.8	8.8	17.2	72.4	17.6	72.9	
$\epsilon_{nx}[\text{mm.mrad}]$	1.0	1.0	57.7	93.4	57.7	91.8	57.6	91.5	
$\epsilon_{nz}[\text{mm.mrad}]$	1.0	1.0	14.8	30.6	14.8	30.4	14.8	28.0	

Discussion: This section seems to come close to a summary/conclusion, but lacks a strong concluding statement.

We thank the referee for this remark and we accordingly change the conclusion for a shorter one. The conclusion :

We have shown that the LWFA electron beam properties can be manipulated through an adequate transport line, mitigating the performance that does not meet that of state-of-the-art conventional accelerators for some specific applications. In particular, we have compensated for beam pointing misalignments, corrected dispersion and adjusted the focused energies via fine tuning of individual gradient strengths and magnetic axis positions of the QUAPEVA triplet. By applying conventional accelerator techniques to laser wake-field acceleration, we have demonstrated the shaping of the electron beam longitudinal and transverse phase-spaces. The undulator synchrotron radiation from such controlled electron beam has been successfully observed after the 8 m manipulation line. Even with the contributions of the remaining shot-to-shot fluctuations, the radiation measurement is interpreted and is in a good agreement with the numerical analysis. This electron beam control is not restricted to LWFA but could be of interest for various advanced accelerating techniques. These results pave the way for further merging of novel and conventional accelerator techniques towards future applications, such as the new generation of colliders, requiring stages of LWFA accelerating "modules" or Free Electron Laser applications, relying on more optimised LWFA performance.

has been modified as:

“In conclusion, we have shown that the LWFA electron beam properties can be manipulated through an adequate transport line, mitigating the performance that do not meet the one of state-of-the-art conventional accelerators for some specific applications. This electron beam control is not restricted to LWFA but could be of interest for various advanced accelerating techniques. These results pave the way for further merging of novel and conventional accelerator techniques towards future applications, such as the new generation of colliders, requiring stages of LWFA accelerating “modules” or Free Electron Laser applications, relying on more optimized LWFA performance that are progressing very fast.”

REVIEWERS' COMMENTS:

Reviewer #1 (Remarks to the Author):

Comments on `Shaping of laser plasma accelerated electrons for light sources` by T. André, I. A. Andriyash, A. Loulergue, et al

The authors have answered well to all my questions. Moreover, improvements done in response to the other Referees make the manuscript worth for publication as it is.

Reviewer #2 (Remarks to the Author):

Authors have addressed all of my comments with the revised manuscript. The quantitative descriptions are easy to follow now. It is not clear if the scheme that increases the beam emittance by up to two orders of magnitude can be considered as a solution to the challenge of re-matching LWFA beams.

I have a few optional suggestions for authors:

1. to use x-y coordinates (and not x-z) for transverse plane;
2. to quantify energy resolution of the slit in the middle of the chicane and compare it to the selection based on chromaticity of triplet focusing (slit at the focal position in X and/or Y).

Reviewer #3 (Remarks to the Author):

I am broadly satisfied with the rewrite. I think that the authors have addressed my questions/comments from review of the initial submission.

Answer to Referee: Control of laser plasma accelerated electrons for light source applications

Reviewer n°1 (Remarks to the Author):

The authors have answered well to all my questions. Moreover, improvements done in response to the other Referees make the manuscript worth for publication as it is.

The authors thank the referee for the positive feedback and are happy for the recommendation to publish in Nature Communications.

Reviewer n°2 (Remarks to the Author):

Authors have addressed all of my comments with the revised manuscript. The quantitative descriptions are easy to follow now. It is not clear if the scheme that increases the beam emittance by up to two orders of magnitude can be considered as a solution to the challenge of re-matching LWFA beams.

The authors agree with the referee concerns about the emittance growth even if it can be mitigated with the initial divergence of the electron beam. Besides, the slice emittance growth in a given range of energies of interest is smaller and can be handled for some applications.

I have a few optional suggestions for authors:

1. to use x-y coordinates (and not x-z) for transverse plane;

The authors prefer to stay with their usual coordinates system.

2. to quantify energy resolution of the slit in the middle of the chicane and compare it to the selection based on chromaticity of triplet focusing (slit at the focal position in X and/or Y).

The authors consider the question opened by the referee. Strong transverse chromaticity of the beam can also be possibly used to sort the particle energies, which might however make beamline more sensitive to the beam pointing variations and require significant modifications of its optics. The authors estimates that this question is a bit out of the work and do not add it to the present manuscript.

Reviewer n°3 (Remarks to the Author):

I am broadly satisfied with the rewrite. I think that the authors have addressed my questions/comments from review of the initial submission.

The authors thank the referee for the positive evaluation of the current version of the manuscript.